# The Role of *Fusobacterium nucleatum* in Oral and Colorectal Carcinogenesis

**DOI:** 10.3390/microorganisms11092358

**Published:** 2023-09-20

**Authors:** Pamela Pignatelli, Federica Nuccio, Adriano Piattelli, Maria Cristina Curia

**Affiliations:** 1COMDINAV DUE, Nave Cavour, Italian Navy, Stazione Navale Mar Grande, 74122 Taranto, Italy; pamela.pignatelli@marina.difesa.it; 2MARICENSELEZ ANCONA, Centro di Selezione M.M., Italian Navy, 60127 Ancona, Italy; federica.nuccio@marina.difesa.it; 3School of Dentistry, Saint Camillus International University for Health Sciences, 00131 Rome, Italy; apiattelli51@gmail.com; 4Facultad de Medicina, UCAM Universidad Católica San Antonio de Murcia, 30107 Murcia, Spain; 5Department of Medical, Oral and Biotechnological Sciences, G. d’Annunzio University of Chieti-Pescara, 66100 Chieti, Italy

**Keywords:** oral microbiota, oral squamous cell carcinoma, colorectal cancer, Wnt/βcatenin, periodontal bacteria, immune response

## Abstract

In recent years, several studies have suggested a strong association of microorganisms with several human cancers. Two periodontopathogenic species in particular have been mentioned frequently: *Fusobacterium nucleatum* (*F. nucleatum*) and *Porphyromonas gingivalis*. Chronic periodontal disease has been reported to be a risk factor for oral squamous cell carcinoma (OSCC), colorectal cancer (CRC) and pancreatic cancer. *F. nucleatum* is a Gram-negative anaerobic bacterium that lives in the oral cavity, urogenital, intestinal and upper digestive tract. It plays a significant role as a co-aggregation factor, with almost all bacterial species that participate in oral plaque formation acting as a bridge between early and late colonizers. *F. nucleatum*, gives an important inflammatory contribution to tumorigenesis progression and is associated with epithelial-derived malignancies, such as OSCC and CRC. *F. nucleatum* produces an adhesion protein, FadA, which binds to VE-cadherin on endothelial cells and to E-cadherins on epithelial cells. The last binding activates oncogenic pathways, such as Wnt/βcatenin, in oral and colorectal carcinogenesis. *F. nucleatum* also affects immune response because its Fap2 protein interacts with an immune receptor named TIGIT present on some T cells and natural killer cells inhibiting immune cells activities. Morover, *F. nucleatum* release outer membrane vesicles (OMVs), which induce the production of proinflammatory cytokines and initiating inflammation. *F. nucleatum* migrates from the oral cavity and reaches the colon hematogenously but it is not known if in the bloodstream it reaches the CRC as free, erythrocyte-bound bacteria or in OMV. *F. nucleatum* abundance in CRC tissue has been inversely correlated with overall survival (OS). The prevention and treatment of periodontal disease through the improvement of oral hygiene should be included in cancer prevention protocols. FadA virulence factors may also serve as novel targets for therapeutic intervention of oral and colorectal cancer.

## 1. Introduction

In the last two decades, significant evidence has emerged implicating bacteria in the aetiology of some cancer types such as *Helicobacter pylori* in non-cardia gastric carcinoma and mucosa-associated lymphoid tissue (MALT) lymphomas, *Chlamydia trachomatis* in cervical cancer and *Salmonella typhi* in gallbladder cancer [1]. Some biotypes of the oral microbiome may contribute more to carcinogenesis than others, and the question arises of whether they become unbalanced as a result of the tumor or in some way they influence the onset of the tumor. Two periopathogenic species in particular have been frequently mentioned: *Fusobacterium nucleatum* (*F. nucleatum*) and *Porphyromonas gingivalis* (*P. gingivalis*). For the first time, Nagy et al. detected significantly higher levels of *Porphyromonas* spp. and *Fusobacterium* spp. in oral squamous cell carcinoma (OSCC) tissues than in adjacent healthy mucosa through swab and culture-based analysis [2].

For *F. nucleatum*, maybe the accumulation of host mutations makes the cells more susceptible to infection. Furthermore, binding of secreted bacterial proteins with host cells might activate oncogenic and immunosuppressive pathways. Previous studies have reported that microbes promote cancer in susceptible hosts, i.e., after the “first hit” has occurred. For instance, in patients affected by familial adenomatous polyposis (FAP), who have a high susceptibility to developing colon cancer, their colons are significantly enriched with Bacteroides fragilis and colibactin-producing E. coli. These two microorganisms cooperate in inducing early tumorigenesis and mortality in Apc min/+ mouse model [3]. 

In 2013 Keku et al. argued whether Fusobacterium colonization was a consequence or a cause of CRC [4]. In 2014 Sears and Garret investigated the complex host–microbiota interactions to understand whether to consider microbes as a primary or secondary cause in the onset of CRC, and whether to associate them with CRC initiation and/or progression of CRC [5].

Dysbiosis, with subsequent bacterial invasion, causes inflammation and inflammation causes cancer through a pro-inflammatory microenvironment that subsequently becomes a tumor microenvironment (TME), which downregulates the adaptive anti-tumor immune response and accelerates the CRC progression [6].

*F. nucleatum* is now considered a cancer-leading bacteria given its ability to stimulate oncogenic pathways through its proteins.

It is also true that host genetics is the presumed initiator permissive to development of a dysbiotic microbiome due to consequent gene–host microbiota interactions. Therefore, the mutations or polymorphisms of the host facilitate the infection of the bacterium. In this case *F. nucleatum* could be a consequence of cancer.

In any case, β-catenin signaling is activated in approximately 90% of CRCs, and both FadA and lipopolysaccharides (LPS) from *F. nucleatum* are the main activators [7,8]. This makes it a pro-carcinogenic bacterium [9].

Chronic periodontal disease is an independent risk factor for oropharyngeal squamous cell carcinoma (OPSCC), for degeneration of potentially malignant oral lesions into OSCC, gastric cancer and pancreatic cancer; indeed, patients with periodontitis had a 2.66-fold higher risk for developing oral cancer.

Colorectal cancer (CRC) is the third most common type of cancer worldwide and the second most frequent cause of cancer death in 2020, with nearly 1 million deaths per year in 2020, leading to almost 1 million deaths per year [10,11]. The increased risk of CRC was associated with a change in fecal microbiota composition: lowered bacterial diversity, decreased Gram-positive, fiber-fermenting *Clostridia* and higher presence of Gram-negative, proinflammatory genera. The risk of CRC increased with elevated *Porphyromonas* (32.1% vs. 16.2% in the case vs. control subjects) and *Fusobacterium* (34.3% vs. 28.1%), assayed by quantitative polymerase chain reaction (qPCR). *Fusobacterium* was significantly more present in case patients than controls (31.9% vs. 11.7%), and was associated with increased CRC risk [12].

## 2. *Fusobacterium nucleatum*

*F. nucleatum* is an obligate anaerobic bacterium that belongs to the Gram-negative group. It is typically found in the mouth, as well as in the urogenital, intestinal and upper digestive systems; however, it is most commonly present in oral plaque.

It is an opportunistic pathogen from the Fusobacteriaceae family, frequently detected in oral and systemic infections. This family includes nine species *F. nucleatum* (subspecies *nucleatum*, *polymorphum*, *vincentii*, *animalis*, *fusiforme*, and *canifelium*), *F. necrophorum* (subspecies *necrophorum* and *funduliforme*), *F. ulcerans*, *F. gonidiaformans*, *F. mortiferum*, *F. naviforme*, *F. necrogenes*, *F. russii* and *F. varium. F. nucleatum* is linked to diverse human illnesses, including periodontitis inflammatory, bowel disease, angina, ulcerative colitis, persistent otitis, sinusitis, peritonsillar abscesses, brain abscesses, pulmonary abscesses, Crohn’s disease, gynecological abscesses, neonatal sepsis, Lemierre’s syndrome and infective endocarditis [13,14,15,16,17].

*F. nucleatum* plays a significant role as co-aggregation’s factor with almost all bacterial species that participate in oral plaque formation [18]. The protagonists are adhesins (Aid1, CmpA, Fap2, FomA, FadA and RadD) which mediate the microbial co-aggregation, the invasion and facilitating the spread of bacteria [19,20,21]. For example, FadA plays a crucial part in fusobacterial attachment and has been identified as the primary adhesin that facilitates the connection of various Gram-positive initial colonizers [14] and enhances fusobacterial adherence to biofilms [15]. It has the ability to bond with the adhesin SpaP of S. mutans to facilitate the co-aggregation of these two bacterial types and advance the organization of biofilms. [21,22,23]. Therefore, within a dental plaque biofilm, *F. nucleatum* plays a vital function as a connecting agent, serving as a bridge organism that links initial colonizers like Streptococcus spp. to the predominantly anaerobic secondary colonizers it can also adhere to, such as *P. gingivalis* and *Aggregatibacter actinomycetemcomitans*, but also yeast like *Candida albicans* [24]. Indeed, another notable adhesin in fusobacterial adhesion is RadD; this facilitates attachment not just to bacteria but also to the yeast Candida albicans, which is likewise a component of the oral microbiota [16,19]. Two additional fusobacterial adhesins, Aid1 and CmpA, have also been linked to this interaction [25]. 

*F.* nucleatum is additionally crucial in periodontitis as it directly molds the host responses and enhances the contagion potential of other pathogens. Periodontitis is a chronic disease with a multifactorial, locally destructive etiology that decreases quality of life [26,27]. In advanced stages, it destroys bone support, mobility and tooth loss by activating the host’s immune and inflammatory response [28]. Bacterial plaque impairs the periodontal epithelium and promotes macrophage chemotaxis, interleukin release and osteoclast activation [29]. *Fusobacterium* is an intermediate colonizer of the oral biofilm, bridging between the initial aerobic colonizers and the late, highly virulent, anaerobic colonizers [30,31]. Specifically, F. nucleatum can stimulate the production of the antimicrobial peptide β-defensin 2, a small cationic peptide and proinflammatory cytokines, including IL-6 and IL-8, regulating the immune response, in the oral epithelium [32]. Such *F. nucleatum*-driven inflammation contributes to disease progression in a model of oral tumorigenesis. In these disease-causing environments, *F. nucleatum* impacts the operation of immune cells, like myeloid cells, wherein it triggers NF-κB activation, leading to the generation of TNF [33].

*F. nucleatum* is very often implicated in extraoral disease, and as other Gram-negative bacteria, release outer membrane vesicles (OMVs) both in vitro and in vivo; these nanoparticles have been linked as significant participants in bacterial disease development [23,24]. OMVs typically contain LPS, deoxyribonucleic acid (DNA), adhesive proteins and enzymes, leading to their suggestion as a carrier system for these disease-causing factors; proteins in *F. nucleatum* OMVs with potential virulence functions are antigenic components like FomA, FadA, FadD, Fad-I, NapA, ClpB, GroEL, TraT and YadA which can trigger Toll-like receptors (TLRs) on epithelial or immune cells. 

TLR stimulation is associated with activation of the NF-κB pathway and induction of proinflammatory cytokine release [25]. Indeed, *F. nucleatum* incites inflammation in normal epithelial cells both in vitro and in vivo. In patients receiving antibiotics, the microbiome fails to confer resistance to colonization, allowing *F. nucleatum* to establish residency. F. nucleatum adheres to the intestinal mucus layer in the context of a modified gut microbiome, utilizing secreted compounds carried within OMVs. These OMVs engage epithelial TLRs, such as TLR4, leading to phosphorylation and activation of ERK, CREB and NF-κB, thereby initiating the synthesis of proinflammatory cytokines and triggering inflammation [34].

A lot of research and experiments demonstrated that *F. nucleatum* has also a significant role in colorectal cancer, influencing many stages of colorectal cancer progression. It can increase cell proliferation in cancer cells [35], modify the tumor microenvironment [36] and block the antitumoral immune responses of natural killer; it is likely that F. nucleatum may also contribute to the cancer’s metastatic dissemination [37].

## 3. Interrelationships between *P. gingivalis* and *F. nucleatum*

*F. nucleatum* is an intermediate colonizer that enables the adhesion of late colonizing bacteria and facilitates their survival by releasing metabolites. Due to this bridging function, it is the most commonly detected bacterium in clinical infections of various body sites [38].

Oral bacteria can become resistant to antiseptics and host defense mechanisms such as phagocytosis, organizing into biofilms.

*P. gingivalis* enhanced the growth of *F. nucleatum* by releasing a molecule other than AI-2, proposed to be a signal mediating message among different species in a mixed-species community and with synergistic effected on biofilm formation [39].

*F. nucleatum* sustained the growth of *P. gingivalis* in oxygenated and carbon dioxide-poor environments [40]. Lactose inhibits aggregations between *F. nucleatum* and late colonizing bacteria and attachment to mammalian cells, including human buccal epithelial cells and gingival and periodontal ligament fibroblasts, by acting on galactose-binding adhesin [41].

*P. gingivalis*, which is highly proteolytic, activates plasminogen bound to *Fusobacterium*, forming bacterial plasmin, a plasma serine protease. *F. nucleatum*, generally non-proteolytic, acquires proteolytic capacity through surface plasmin, being able to process potential peptide signals in the community [42]. The peptides produced can be used as nutrients by fusobacteria or other bacteria in the biofilm.

Co-infection of *F. nucleatum* and *P.* gingivalis could protect the couple from the host immune response by decreasing β-defensin 2 production [43].

Polymicrobial infections can increase the virulence of individual bacteria; in fact, *F. nucleatum* significantly increased the adhesion and invasion abilities and efficiency of *P. gingivalis* and *A. actinomycetemcomitans.* This enhanced adhesion and invasion of *P. gingivalis* and *A. actinomycetemcomitans* boosted by *Fusobacterium* could be inhibited by galactose [44].

The interactions between orange *(P. micros*, *P. nigrescens. P. intermedia*, *F. nucleatum*, *F. periodonticum*) and red (*P. gingivalis*, *T. forsythia*, *T. denticola*) complex bacteria may have a synergistic impact on immune regulation on epithelial cells. Socransky classified the bacteria associated with periodontal disease into a scheme of microbial succession followed by reciprocal host–bacterial interaction [45]. Each bacterial complex was color coded. The yellow, green and purple clusters comprise the early colonizers of dental plaque, while the orange and red clusters are the late colonizers. Those classified as red are considered the most dangerous and responsible for habitat change, manifested clinically as gingivitis in the early and periodontitis in the more advanced stages [46].

## 4. Mechanisms of Carcinogenesis of *F. nucleatum*

CRC is a common cancer worldwide and is essentially a genetic disease; however, it is still unknown precisely what events contribute to precipitate the initial disease and promote progression. *F. nucleatum* is found in human tumors and has been associated with CRC since the past decade [37,47,48]. *F. nucleatum* has been found in most CRC tissues, mainly in the proximal colon [49,50,51]. *F. nucleatum* may move from the oral cavity to distant body sites via the hematogenous route [52]. It has been found to be enriched in the microbiota adherent to the mucosa and to have the ability to adhere to and invade human epithelial and endothelial cells [53]. It is considered a pro-carcinogenic bacterium due to its involvement in various stages of CRC. *F. nucleatum* and *P. gingivalis* are the most characterized bacteria regarding their pro-inflammatory and oncogenic role. Both bacteria enter human epithelial and endothelial cells, establishing persistent intracellular infections, spreading beyond the oral cavity [54] and finally playing their pathogenetic role. For *F. nucleatum*, it depends on its FadA adhesion protein [55,56]. The FadA protein is present in two forms: the first, named pre-FadA, consists of 129 amino and is attached to the membrane and the second, mature and secreted, is named FadA (mFadA) and consists of 111 amino acids [57]. The two forms combine in an activity complex, FadAc, which is internalized, anchored in the inner membrane and protrudes through the outer membrane. The formation and internalization of the activity complex ensures invasion of the bacteria into the host cells [57].

FadA was originally reported to bind on endothelial cells to vascular endothelial (VE)-cadherin [55]. Cadherins belong to a family of cell adhesion glycoproteins, whose structure consists of five extracellular domains (EC1-EC5), a transmembrane domain and a cytoplasmic tail that binds various cytoplasmic proteins including β-catenin [58]. FadA binds to the host endothelial cell by the EC5 receptor and after its binding it invades endothelial cells. Given the 33.5% similarity between VE- and epithelial (E)-Cadherins, it was speculated that FadA also bound to E-cadherin. In fact, in the host epithelial cell, FadA binds to the EC5 domain of the cell adhesion molecule E-cadherin [59], a strong tumor suppressor that inhibits tumor growth and development [60].

There are two pathways activated by *F. nucleatum* FadA recognized in colon carcinogenesis. In one pathway the FadA adheres to and invades human epithelial cells and endothelial cells. This invasion is required for the stimulation of inflammation and leads to an increase of inflammatory cytokines, particularly interleukin-8 (IL-8), which is regulated by the p38 MAPK signaling pathway, interleukin-10 (IL-10), tumor necrosis factor-a (TNF- a) and nuclear factor kappaB (NF-κB) [35,61,62,63]. This at first generates a proinflammatory microenvironment that subsequently becomes a tumor microenvironment which downregulates the adaptive anti-tumor immune response and accelerates the CRC progression. 

In the other pathway, the intestinal barrier impairment caused by mutations in epithelial cells gives *F. nucleatum* the opportunity for adherence and subsequent invasion into cells. FadA interacts with the host E-cadherin on the epithelial cell (Figure 1) and this binding requires both the intact pre-FadA and mFadA without the signal peptide. This interaction favors the internalization of FadA by epithelial cells and the activation of the β-catenin-regulated transcription (CRT). Activated β-catenin enters the cell nucleus from cytoplasm, increases the expression of wnt signaling genes such as wnt7a, wnt7b and wnt9a, the myc and cyclin D1 oncogenes, the transcription factors such as the lymphoid enhancer factor (LEF-1) and the T-cell factor such as TCF1, TCF3 and TCF4. It also promotes NF-κB gene expression, pro-inflammatory genes and the expression of many other oncogenes [35].

E-cadherin can be internalized via clathrin [60]. Pitstop2, a clathrin inhibitor [64], prevented *F. nucleatum* invasion but without affecting attachment. The following stimulation of the inflammatory gene NF-κB and of the cytokines IL-6, 8 and 18 is abolished in the presence of the clathrin inhibitor. So, *F. nucleatum* cell invasion via internalization of E-cadherin by clathrin and subsequent stimulation of expression of inflammatory genes could represent an additional mechanism of carcinogenesis [35]. 

The work by Rubinstein et al. on CRC cell lines and mice was very important to better understand the relationship between *F. nucleatum* and the Wnt/β-catenin pathway in the colorectal carcinogenesis [59]. Annexin A1 (ANXA1), a Ca^2+^-dependent phospholipid-binding proteins belonging to the annexin family [65] has been reported with increased expression in proliferating CRC cells. The authors found that *F. nucleatum* selectively incites the growth of colorectal cancer cells through activation of Annexin A1. Using Western blot analysis, the authors highlighted an increase of expression of E-cadherin, Annexin A1 and β-catenin proteins as binding to FadA, suggesting possible interactions in a multi-component complex. 

Co-localization of FadAc with E-cadherin and Annexin A1 both on the cell membrane and inside the cell was confirmed by confocal microscopy analysis. This is consistent with the previous findings about the binding of FadAc with vascular VE-cadherin and subsequent internalization [35,55]. Annexin A1 is a growth factor of CRC and required for activation of Wnt/β-catenin signaling [66,67].

In particular, it was defined as a Wnt/β-catenin modulator. The authors hypothesize that binding FadA to E-cadherin engages Annexin A1 to form on the membrane a more stable complex, in turn reducing cytoplasmic amount of Annexin A1. The authors also proposed a “two-hit” model, in which the increased expression of Annexin A1 is the first “hit” and *F. nucleatum* is the second “hit”. This model extends the known “adenoma-carcinoma” model [68] in identifying microbes as facilitators of cancer progression because they intervene only after the benign cells progress to a malignant phenotype. Finally, several potential mechanisms by which *F. nucleatum* promotes colorectal carcinogenesis are proposed: (a) the accumulation of host driver mutation(s), for example in APC gene, or increased expression of Annexin A1, make the cells more susceptible to *F. nucleatum* infection and to FadA attack with subsequent invasion; (b) the binding of FadA with E-cadherin on the cell membrane causes the latter to lose its tumor-suppressor activity and as a consequence β-catenin signaling is activated and (c) *F. nucleatum* stimulates the binding of Wnt ligands to frizzled with consequent decrease in β-catenin phosphorylation and its translocation into the nucleus where it stimulates the over-expression of oncogenes [69] (Figure 1). Furthermore, *F. nucleatum* is involved in different CRC stages, in particular during tumor progression, metastasis and chemoresistance. Its role is manifested in reprogramming the TME by interacting with cancer and immune cells [9] confirming that the intratumor microbiota is a part of TME [70,71].

At present, due to the many involvements of Wnt/β-catenin signaling in the cellular functions, only a few drugs acting as Wnt inhibitors have been approved by the FDA for the treatment of cancer [72]. Also, E-cadherin is a difficult therapeutic target since it is ubiquitarian. On the contrary, Annexin A1 is a promising therapeutic target to inhibit Wnt/β-catenin signaling due to its increased and selective expression in proliferating cancerous cells; this would avoid adverse side effects. It has been reported that *F. nucleatum* and Annexin A1 cause chemo-resistance metastasis and poor prognosis in CRC [12,51,73,74,75]; therefore, Annexin 1 can be considered a biomarker for all cancers in which *F. nucleatum* is present. 

TLR4 and p21-activated kinase 1 (PAK1) could also be potential pharmaceutical targets for the treatment of *F. nucleatum*-related CRCs, given the hypothesis that the bacterium in CRCs activates β-catenin signaling through the TLR4/PAK1 cascade [7].

A second virulence factor, an autotransporter protein, Fap2, has been shown to potentiate the progress of CRC via inhibiting immune cell activity [76].

## 5. *F. nucleatum* and Immune Response

A particular characteristic of the microbiota is its ability to alter the antitumor immune response and several studies have documented this aspect [77]. Interestingly, in the paper by Zitvogel et al., the authors argue that there is a triangular relationship in the tumor that contributes to the immune control or escape of distant tumors. This triangulation, consisting of the microbiome, the immune system and the cancer, on one hand can activate antitumoral immune effectors and on the other hand can stimulate immunosuppressive immune populations. These different mechanisms occur due to microbiota-derived pathogen-associated molecular patterns (PAMPs), danger-associated molecular patterns (DAMPs) and cytokines and chemokines present in the tumor microenvironment.

*F. nucleatum* produces a protein that in CRC blocks a receptor on T cells thus inhibiting their cytotoxic activity on tumor cells [50,76] (Figure 2). Gur et al. have demonstrated that *F. nucleatum* creates a tumor microenvironment with immunosuppressive activity that contributes to CRC progression. The authors also demonstrated that tumor-infiltrating lymphocytes (TIL) expressed an immune receptor named T-cell immunoreceptor with Ig and ITIM domains (TIGIT), present on some T and NK cells which regulates T-cell-mediated immunity. TIGIT has been shown to be over expressed on CD8+ TILs from individuals with cancer [78].

*F. nucleatum* affects human NK cell cytotoxicity because its Fap2 protein interacts with TIGIT [76]. In this mechanism, Fap2 creates a bacterium-dependent, tumor immune evasion inhibiting T-cell activities via TiGIT. Normally the presence of TILs in tumors is associated with better clinical outcomes because they are implicated in killing tumor cells. By means of this mechanism, *F. nucleatum* creates an escape of tumor cells. In addition, the anoxic and acidic tumor microenvironment would be more suitable for Fusobacterium life. Furthermore, the metabolites of Fusobacterium enroll CD11b+ myeloid derived suppressive cells (MDSCs), usually highly abundant in cancer (Figure 2). They are reported to be more abundant in tumor tissues of Fusobacterium-fed APC^min^ mice, in turn suppressing anti-tumor immunity and promoting CRC carcinogenesis [47]. Due to their immunosuppressive activities, tumor tissue with high infiltration of MDSCs may predict poor prognosis and drug resistance [79].

*F. nucleatum* is associated to epithelial-derived malignancies, such as OSCC, colorectal cancer (CRC) and breast cancer [47,48,80,81,82].

## 6. *F. nucleatum* in Oral and Colorectal Cancer

Tumor and healthy tissue had different bacterial species at the outer surface or in the tissues; most of those in the tumor tissue were saccharolytic and aciduric species. This could indicate a change in the deep tumor microenvironment leading to a selection of ascarolytic or weakly fermentative species. *Fusobacterium* and *Prevotella* were isolated from oral cancer samples and they were able to survive at relatively low pH in hypoxic environments [83]. The bacterial composition of the deep tissues was similar but less differentiated than the overlying mucosa and tends to imply a local origin for the bacteria found within the tumor. Quite different was the microbiota between tumor and control tissues demonstrating a specificity of the microenvironment in the two conditions. The results obtained with culture techniques were confirmed by the introduction of 16s rRNA sequencing or next generation sequencing (NGS) [2,84].

Key studies about the role of *F. nucleatum* in the carcinogenesis processes of CRC and OSCC are summarized in Table 1.

Pushalkar et al. reported the relative abundance of bacteria in tumor samples was significantly dissimilar to adjacent normal mucosa; OSCC tumors and adjacent nontumor mucosa had no phylogenetic differences [85].

**Table 1 microorganisms-11-02358-t001:** Summary of key studies about the role of *F. nucleatum* in the carcinogenesis processes of CRC and OSCC. CC: colon cancer; CRA: colorectal adenoma; CRC: colorectal cancer; *Fn: Fusobacterium Nucleatum*; HNSC: head and neck squamous cell carcinoma and OSCC: oral squamous cell carcinoma.

Reference	Topic	Type of Study	Conclusion
Rubinstein, M.R., 2013 [59]	Molecular mechanisms	In vivo/Animal	FadA of *Fn* binds to E-cadherin on normal and CRC cells, promoting their attachment and invasion. FadA levels in adenomas and adenocarcinomas are >10–100-fold higher than in normal subjects. Its increased expression correlates with CRC cell growth and oncogenic and inflammatory responses.
Castellarin, M., 2012 [37]	Molecular mechanisms	In vitro	*Fusobacterium* was higher relative abundance in tumors with >50% circumferential involvement. High *Fusobacterium* tumors were significantly more likely to have regional lymph node metastases.
Kostic, A.D., 2013 [47]	Molecular mechanisms	In vivo/Animal	*Fn* were enriched in human colonic adenomas and in stool samples from CRC patients compared to healthy subjects. It generated a pro-inflammatory microenvironment via the recruitment of tumor-infiltrating myeloid cells and the expression of pro-inflammatory genes.
Abed, J., 2020 [52]	Molecular mechanisms	In vivo/Animal	Fn can colonize the CRC by migrating from the oral microbial reservoir through the hematogenous route.
Gur, C., 2015 [76]	Immunity	In vitro	The Fap2 protein of *Fn* inhibited immune cell activity, tumor-infiltrating lymphocytes expressed TIGIT and T-cell activities.
Zhao, H., 2017 [86]	OSCC carcinogenesis	In vivo/Human	A group of periodontitis-correlated taxa, including *Fusobacterium*, was found to be significantly enriched in OSCC samples. The operational taxonomic units (OTUs) associated with *Fusobacterium* were highly involved in OSCC and demonstrated good diagnostic power.
Yost, S.,2018 [87]	OSCC carcinogenesis	In vivo/Human	*Fn* was the most active bacterium expressing putative virulence factors in the tumor sites. At tumor sites proteolysis, DNA mismatch repair, carbohydrate metabolism, cell redox homeostasis and citrate transport were all over represented.
Yang, C.-Y., 2018 [88]	OSCC progression	In vivo/Human	The abundance of *Fusobacteria* increased significantly with the evolution of oral cancer from the healthy controls to OSCC stage 1 through stage 4.
Desai, S., 2022 [89]	OSCC prognosis	In vivo/Human	*Fusobacterium* presence is associated with an inflamed, innate immune cell-enriched and pro-tumorigenic microenvironment, as opposed to the HPV-positive HNSC tumors. *Fn* is also associated with poor prognosis, nodal metastases and high extracapsular spread in tongue tumors.
Neuzillet, C., 2021 [90]	OSCC prognosis	In vivo/Human	*Fn*-associated OSCC had a specific immune microenvironment, was more frequent in older, non-drinking patients, with a favorable prognosis. Patients with OSCC had significantly longer overall survival, relapse-free and metastasis-free survival.
McCoy, A.N., 2013 [61]	CRC carcinogenesis	In vivo/Human	There was a higher abundance of *Fusobacterium* species in the sigmoid than right side CRC location. *Fn* was more abundant in CRA than controls. CRA but not controls had a significant positive correlation between local cytokine gene expression and *Fusobacterium* quantity.
Rezasoltani, S., 2018 [91]	CRC carcinogenesis	In vivo/Human	Higher numbers of Fn were detected in adenomatous polyps in contrast to samples from the normal cases, hyperplastic polyps and sessile serrated polyps.
Komiya, Y., 2019 [92]	CRC carcinogenesis	In vivo/Human	Identical *Fn* strains were detected in both CRC and saliva from 42.9% of the patients. *Fn* was detected from stages 0 to IV and there were no significant differences in the detection rate among each lesion site.
Ito, M., 2015 [93]	CRC carcinogenesis	In vivo/Human	*Fn* was significantly higher in CRCs than in premalignant lesions of any histological type, in the latter increased gradually from sigmoid colon to cecum, frequently associated with CIMP-high lesions. *Fn* positivity increased according to histological grade.
Genua, F., 2023 [94]	Serum Antibody in CRA and CRC	In vivo/Human	IgG sero-positivity to *Fn* was associated with an increased CRC risk. *Fn* abundance in the normal mucosa positively correlated with the IgA response to the *Fn* antigen.
Lee, J.B., 2021 [95]	CC prognosis	In vivo/Human	*Fn* enriched right-sided metastatic, and recurrent colon cancer was significantly associated with worse progression-free survival, indicating that *Fn* enriched right-sided colon responded less to palliative cytotoxic chemotherapy.
Mima, K., 2016 [50]	CRC prognosis	In vivo/Human	The quantity of *Fn* DNA in colorectal cancer tissue was positively associated with pT stage and with colorectal cancer-specific mortality, independent of clinical, pathological, and major tumor molecular features.
Pignatelli, P., 2021 [81]	CC prognosis	In vivo/ Human	*Fn* oral concentration influenced colon tissue concentrations. *Fn* was statistically significantly higher in pathological tissue compared to the matched adjacent non-neoplastic mucosa. The *Fn* quantity in the colon cancer tissue predicted the staging.

In the study by Schmidt et al., the abundance of the genus *Fusobacterium* was significantly higher in the tumor samples compared with contra-lateral normal; furthermore, reducing the abundance of *Streptococcus* species resulted in an enhanced proinflammatory environment, since it has been reported that it attenuates *F. nucleatum* capable of increasing pro-inflammatory responses of oral epithelial cells [96]. Zhao et al. identified drastic changes in the surface bacterial communities of the OSCC group, *Fusobacterium* species were significantly increased, while *P. gingivalis* did not differ in abundance between the OSCC and healthy groups [86].

Yost et al. used metatranscriptomic analysis to characterize bacterial functional activities, broad gene expression profiles based on the set of transcripts synthesized by the microbiota, in OSCC sites, adjacent non-tumor-affected sites of OSCC subjects and healthy controls. *F. nucleatum* was the most active bacterium in expressing putative virulence factors in tumor sites compared with adjacent non-tumor-affected sites and healthy controls. *Fusobacteria* were the best biomarkers for tumor sites based on expression analysis. Its abundance was associated with increased proteolysis, DNA mismatch repair, carbohydrate metabolism, cell redox homeostasis and citrate transport at the tumor site compared with control or adjacent buccal sites in patients with OSCC [87]. Furthermore, genes involved in bacterial chemotaxis, flagellar assembly and LPS biosynthesis were significantly increased the in the OSCC group [82].

High presence of *F. nucleatum* was detected in oral precancerous lesions, proposing a dynamic change of oral microbiota in the progression of oral cancer. *Fusobacterium*, *Leptotrichia*, *Campylobacter* and *Rothia* species were significantly increased in oral leukoplakia (OLK) versus the corresponding healthy contralateral samples, from the analysis of oral swabs,. The co-presence of Fusobacteria and Campylobacter spp. on OLK, as well as in colic lesions, may promote malignant transformation processes [97]. The abundance of *Fusobacteria* increased significantly progressively from tissues of healthy controls (2.98%) to stage 1 OSCC (4.35%) to stage 4 (7.92%). Specifically, *F. periodonticum* increased from stage 1, to stages 2 and 3 and then to stage 4 of OSCC (1.66%, 2.41% and 3.31%, respectively) in oral rinse samples [88].

In murine model of chronic experimental periodontitis induced by *P. gingivalis* and *F. nuclaetum* associated oral tumorigenesis, chronic infection could promote growth of tongue tumors from infected mice, 2.5 times larger, and invasiveness, augmented signaling along the IL-6-STAT3 axis. In addition, cyclin D1 oncogene expression was significantly increased in both the cancerous tongue epithelium and adjacent healthy tissue in infected mice compared with uninfected mice [98]. HPV-positive HNSC form a distinct clinical and molecular oropharyngeal tumor with better prognosis and response to treatment. High *Fusobacterium* tumors represent another distinct subgroup of HNSC, often associated with HPV-positive HNSC. Data on the prognosis, extracapsular spread and lymph node metastasis of *F. nucleatum*-positive tumor are conflicting. Desai et al. reported its association with poor prognosis and lymph node involvement in tongue cancer. This tumor is not associated with alcohol consumption, tobacco smoking or chewing status, but is associated with downregulation of 34 genes, 118 upregulated inflammatory pathway genes, high levels of pro-inflammatory cytokines IL1β, IL8 and IL6, suggesting an inflamed tumor microenvironment. *Fusobacterium* might play a role in lymph node metastasis in oral tongue HPV-positive tumors through the induction of chronic inflammation [89]. In contrast, several studies demonstrated that *F. nucleatum*-positive OSCC, which is more frequent in elderly, non-drinking patients, was associated with lower lymph node invasion and distant recurrence, overall survival (OS), disease-free survival (DFS), cancer-specific survival (CSS) and lower recurrence rate. There was an association of low stage pT or pN with *F. nucleatum* positivity. Probably *F. nucleatum* could promote a “permissive” tumor microenvironment with low TLR4 signaling, low OX40 ligand release, low M2 and CD4 lymphocyte recruitment, despite high levels of pro-inflammatory cytokines [90,99].

The titers of serum antibodies against *P. gingivalis* and *F. nucleatum* were strongly correlated in OSCC patients, but only serum levels of *P. gingivalis* IgG were significantly higher than in healthy controls. Changes in bacterial species and especially increase in *P. gingivalis* and *F. nucleatum* could be used as biomarkers of malignant transformation in oral potentially malignant disorders (OPMD) [100]. Microbial patterns in both tissue and oral rinse samples showed high sensitivity and specificity as novel biomarkers to complement cancer diagnostics in HNSCC. The incorporation of the oral microbiome with salivary tumor biomarkers such as DNA methylation and miRNA expression may support the molecular diagnosis for oral and oropharyngeal cancers. The study of the oral microbiota in other than diagnostic applications suggests experimental adjuvant anticancer treatments. Indeed, new oral cancer therapies could target the virulence factors FimA *of P. gingivalis* and FadA of *F. nucleatum* [101]. Prevention and treatment of periodontal disease, by improving oral hygiene, should be included in cancer prevention protocols. 

Several studies have reported that there was a significant positive correlation between high abundance of *F. nucleatum* and the prevalence of colorectal adenomas, precursors of colorectal cancer [61,91,102]. *F. nucleatum* strains were detected in both CRC tissue, apparently healthy adjacent colonic mucosa and saliva of CRC patients [81,92]. *Fusobacterium* was localized in the mucus layer above the epithelium as well as within the colonic crypts by fluorescence in situ hybridization (FISH) in the colon. FadA gene levels gradually increased from healthy colon tissues to adenoma, and from adenoma to CRC. Levels of FadA in healthy tissues adjacent to adenoma and carcinoma were higher than those in tissues of healthy individuals. This was indicative of field cancerization. There was a strong positive correlation between abundance of *Fusobacterium* species and local concentrations of cytokines IL-10, IL-12 in adenoma, but not in controls. Therefore, *Fusobacterium* may be invasive and may contribute to increased mucosal inflammation in adenoma subjects [61]. It was present in all histological types of premalignant colorectal lesions, acting in the early stage of tumorigenesis. Its abundance was significantly associated with CIMP-high status, larger tumor size and increased gradually from the sigmoid colon to the cecum in sessile serrated adenomas (SSAs) [93]. In colic biopsies from patients with CRC, 10-fold higher levels of the FadA gene were assayed than in normal individuals [37,103]. *F. nucleatum* colic infection induced mucosal and systemic antibody responses, in particular, CRC patients infected with *F. nucleatum* produced higher levels of IgA and IgG than control groups [104]. However, there was not any association of *F. nucleatum* sero-positivity with detection of polyps or advanced adenomas [94]. *F. nucleatum* migrates from the oral cavity and reaches the colon either hematogenously or via the gastrointestinal tract. Hematogenous spread of oral *Fusobacteria* to CRC appears to be the most biologically plausible hypothesis, in fact, Abed et al. demonstrated that the intravenous route for CRC colonization by oral fusobacteria was more efficient than a gastrointestinal route in mouse models of orthotropic CRC. Transient bacteremia is common during dental procedures (tooth extraction, periodontal surgery, scaling and root planning) and during daily brushing and flossing. But it is not known if *Fusobacteria* in the bloodstream reach the CRC as free, erythrocyte-bound bacteria or in OMV. In addition, travel in the bloodstream allows *Fusobacteria* to escape competition with the endogenous colonic microbiota [52]. The microbiota is an intrinsic component of the tumor microenvironment, which can migrate together with primary tumor cells as part of the colonization of metastatic tissue. The same *Fusobacterium* species has been found in distant metastases and paired primary human colorectal tumors [95]. In the presence of *F. nucleatum* not only did CRC exhibit more aggressive but also was more likely to develop recurrence [51]. Treatment with the antibiotic metronidazole, by decreasing *Fusobacterium* load, reduced tumor cell proliferation and tumor growth in mice [12].

The increase of *F. nucleatum* in colic cancer tissue was associated with worse tumor staging, otherwise there was no correlation with grading; there was a trend of correlation between abundance of *F. nucleatum* in non-neoplastic intestinal tissue and staging [81].

*P. gingivalis* and *F. nucleatum* can predict poor OS, DFS, and CSS in cancer patients [105] and high numbers of *F. nucleatum* in CRC tissue is inversely-correlated with OS [74].

## 7. Conclusions

The influence of *F. nucleatum* in the carcinogenic process is undoubted.

We need to understand whether the microbiome triggers the initiation or the progression of oral and colonic carcinogenesis. It remains unknown which bacterial activities influence the tumor-initiating mutation or incite subsequent disease progression. The host-activated signaling pathways and the virulence factors differ between microbes and cancer types, but the predisposed host represents the common factor in microbe-driven carcinogenesis.

The interactions between microbiota and host are influenced by host genetic polymorphisms that influence the stability of the genome and modify immune and metabolic responses. Driver gene mutations that cause local intestinal barrier impairment but also (a) binding of FadA to E-cadherin on the surface of the epithelial cell, (b) increased expression of Annexin A1 and c) binding of Wnt ligands to their receptors, at first activate oncogenic signaling. Then, the metabolites of aggregated *F. nucleatum* Fap2, giving epithelial cells the ability to gather immune cells such as MDSC or altering their activity, such as for NK cells, produce a tumor immunosuppressive microenvironment that promotes progression in oral and colorectal cancer. Moreover, the anoxic and acidic tumor microenvironment renders the reproduction of Fusobacterium more suitable.

*F. nucleatum* was present in all histological types of premalignant oral and colorectal lesions, acting in the early stage of tumorigenesis. In colon FadA, gene levels gradually increased from healthy tissues to adenoma and from adenoma to CRC, indicating a field cancerization. So FadA represents the best biomarker for a therapeutic approach.

Treatment with the antibiotic metronidazole, by decreasing *Fusobacterium* load, reduced tumor cell proliferation and tumor growth in mice.

## Figures and Tables

**Figure 1 microorganisms-11-02358-f001:**
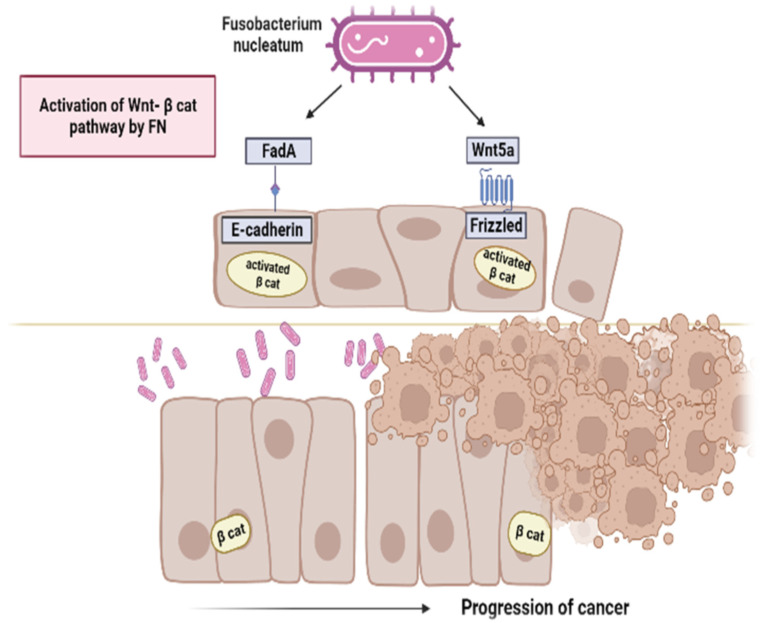
Activation of Wnt/β-catenin pathway by *F. nucleatum*. *F. nucleatum* activates the oncogenic Wnt pathway by binding of FadA with E-cadherin. In OSCC occurrence and progression, *F. nucleatum* is reported to stimulate the binding of Wnt5a with its receptor Frizzled, thereby activating β-catenin. Image created with BioRender (https://biorender.com; last accessed on 28 July 2023).

**Figure 2 microorganisms-11-02358-f002:**
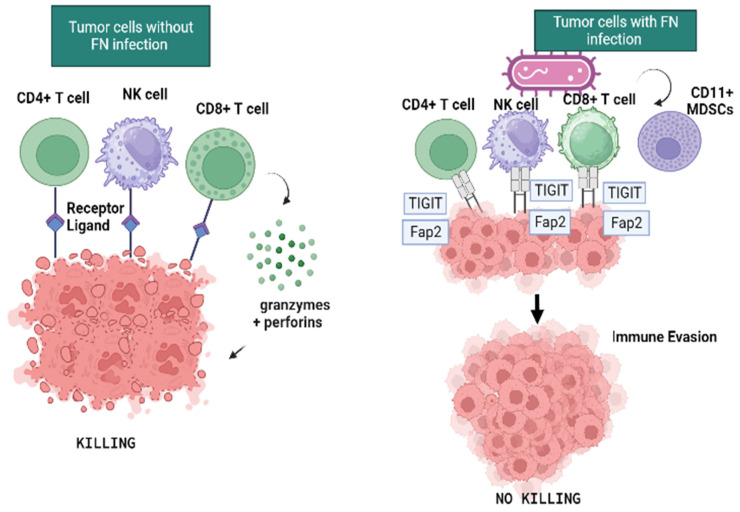
*F. nucleatum* in the tumor microenvironment. In tumor cells without *F.nucleatum* infection, TILs are able to recognize and eliminate tumor cells also with the help of cytotoxic granules released containing granzymes and perforins. In the presence of *F.nucleatum*, its Fap2 protein specifically targets the inhibitory receptor TIGIT, thus protecting tumor cells from immune cell attack. Image created with BioRender (https://biorender.com; last accessed on 28 July 2023).

## Data Availability

Not applicable.

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
