# Peer review of "The Role of *Fusobacterium nucleatum* in Oral and Colorectal Carcinogenesis"

_microorganisms, 2023, doi:10.3390/microorganisms11092358_

Round 1
Reviewer 1 Report
Overall, the review article is well-structured and provides detailed information about the role of Fusobacterium nucleatum in oral and colorectal carcinogenesis, based on significant studies in the literature. However, a suggestion to further enhance the article would be the inclusion of a table summarizing the key studies discussed along with their distinct conclusions. The addition of this table would facilitate the search for specific information for interested readers, offering a comprehensive overview of the findings from each study. This would contribute to a quicker and more efficient understanding of the various perspectives presented in the literature.
In addition, it would be beneficial to include a dedicated methodology section that elucidates the process by which the review was conducted. This section should encompass the databases accessed and the specific search strategies employed to gather relevant studies for this review. Such an inclusion would not only enhance the transparency of the study but also provide readers with a clear understanding of the systematic approach taken to select and analyze the literature.
Author Response
Comments and Suggestions for Authors
Overall, the review article is well-structured and provides detailed information about the role of Fusobacterium nucleatum in oral and colorectal carcinogenesis, based on significant studies in the literature. However, a suggestion to further enhance the article would be the inclusion of a table summarizing the key studies discussed along with their distinct conclusions. The addition of this table would facilitate the search for specific information for interested readers, offering a comprehensive overview of the findings from each study. This would contribute to a quicker and more efficient understanding of the various perspectives presented in the literature.
In addition, it would be beneficial to include a dedicated methodology section that elucidates the process by which the review was conducted. This section should encompass the databases accessed and the specific search strategies employed to gather relevant studies for this review. Such an inclusion would not only enhance the transparency of the study but also provide readers with a clear understanding of the systematic approach taken to select and analyze the literature.
We thank the reviewer for his valuable suggestions. We have added as recommended the summary table. The review was designed as a critical review of the literature on the role of Fusobacterium in the processes of carcinogenesis, as a critical revision it does not include a methodology section. Articles were selected from pubmed and scopus but without following the flow chart of a systematic review.
Reviewer 2 Report
In the manuscript entitled “The role of Fusobacterium nucleatum in oral and colorectal carcinogenesis” the authors aimed to critically review the role of the periopathogen Fusobacterium nucleatum specifically addressing its potential role in mucous membrane carcinogenesis. The topic is very interesting and relevant to the current literature. The manuscript is well-written and the images are well presented. Here are some tips to improve the quality of the manuscript:
INTRODUCTION
_ Line 47: “OSCC” is cited here for the first time but it has been specified in line 51.
_ This paragraph should answer the question: "Why submit a review on this topic"? What is the main object of debate in the literature (short hint)? The authors, only if they deem it useful, could for example cite the debate on the possible causal role of these pathogens: are they unbalanced as a consequence of the presence of a tumor or their a priori alteration favors the appearance of a neoformation?
PARAGRAPH 2
_ Line 94: The authors correctly cited the role of F. nucleatum in periodontitis. However, considering that the manuscript could be read by non-expert colleagues on this topic, a brief description of what periodontitis is, the etiopathogenesis and the specific role of F. nucleatum in dysbiosis should be provided.
PARAGRAPH 3
_ Lines 152-153: again, the manuscript could be read by non-expert colleagues on this topic. Briefly explain what orange and red complexes are.
PARAGRAPH 4
_ Lines 233-234: This consideration is central to the manuscript. Are F. nucleatum and dysbiosis etiological agents or are they simply factors involved in the subsequent tumor pathogenesis? It would be really useful to see this aspect better explored throughout the manuscript.
PARAGRAPH 5
_ Lines 297-298: This sentence should be better connected to the previous speech.
PARAGRAPH 6
_ Line 299: I suggest changing the title of the paragraph in: "F. nucleatum in oral and colorectal cancer".
Author Response
Comments and Suggestions for Authors
In the manuscript entitled “The role of Fusobacterium nucleatum in oral and colorectal carcinogenesis” the authors aimed to critically review the role of the periopathogen Fusobacterium nucleatum specifically addressing its potential role in mucous membrane carcinogenesis. The topic is very interesting and relevant to the current literature. The manuscript is well-written and the images are well presented.
We thank the reviewer for his valuable suggestions.
Here are some tips to improve the quality of the manuscript:
INTRODUCTION
_ Line 47: “OSCC” is cited here for the first time but it has been specified in line 51.
We have corrected it.
_ This paragraph should answer the question: "Why submit a review on this topic"? What is the main object of debate in the literature (short hint)? The authors, only if they deem it useful, could for example cite the debate on the possible causal role of these pathogens: are they unbalanced as a consequence of the presence of a tumor or their a priori alteration favors the appearance of a neoformation?
We have added in the Introduction a few sentences regarding the topic suggested by the reviewer (highlighted in the text) and cited some additional references.
PARAGRAPH 2
_ Line 94: The authors correctly cited the role of F. nucleatum in periodontitis. However, considering that the manuscript could be read by non-expert colleagues on this topic, a brief description of what periodontitis is, the etiopathogenesis and the specific role of F. nucleatum in dysbiosis should be provided.
We have added a few sentences for better explain the topic as suggested by the reviewer (highlighted in the text).
PARAGRAPH 3
_ Lines 152-153: again, the manuscript could be read by non-expert colleagues on this topic. Briefly explain what orange and red complexes are.
We have added a few sentences for better explain the topic as suggested by the reviewer (highlighted in the text).
PARAGRAPH 4
_ Lines 233-234: This consideration is central to the manuscript. Are F. nucleatum and dysbiosis etiological agents or are they simply factors involved in the subsequent tumor pathogenesis? It would be really useful to see this aspect better explored throughout the manuscript.
We have added an additional sentence to better explore this aspect (highlighted in the text).
PARAGRAPH 5 6
_ Lines 297-298: This sentence should be better connected to the previous speech.
We modified the sentence according to reviewer suggestion (highlighted in the text).
PARAGRAPH 6
_ Line 299: I suggest changing the title of the paragraph in: "F. nucleatum in oral and colorectal cancer".
We modified it according to reviewer suggestion.
Round 2
Reviewer 2 Report
In their revised manuscript the authors have addressed all my suggestions. The paper may be considered in this form for publication.